# Guideline Directed Medical Therapy at Discharge and Further Uptitration Leading to Reduction in Indication for Prophylactic ICD Implantation during Protected Waiting Period

**DOI:** 10.3390/jcm11206122

**Published:** 2022-10-18

**Authors:** Elias Waezsada, Julie Hutter, Patrick Kahle, Joerg Yogarajah, Johannes Sperzel, Malte Kuniss, Thomas Neumann, Horst Esser, Christian Hamm, Andreas Hain

**Affiliations:** 1Kerckhoff Heart Center, 61231 Bad Nauheim, Germany; 2Zoll CMS GmbH, 50996 Koeln, Germany

**Keywords:** heart failure, GDMT, wearable cardioverter/defibrillator, beta blocker, ACE-inhibitors, ARNI, MRA, sudden cardiac death, ICM, NICM

## Abstract

Heart failure with reduced ejection fraction (LV-EF < 35%) is diagnosed in app. 11,000,000 patients worldwide. For the treatment of these patients, guideline directed medical therapy has proven to reduce mortality and rehospitalization regardless of the disease’s etiology. It is implemented to treat clinical symptoms by improving the left ventricular ejection fraction. Patients with a transient risk of ventricular tachycardia and sudden cardiac death can be protected by a defibrillator vest. The defibrillator vest is capable to detect and terminate ventricular arrhythmias during Guideline Directed Medical Therapy (GDMT). It is based on the recommendations of the European society of cardiology for 3 months. Afterwards, the WCD wear time could be prolonged, or, in case of persistent low ejection fraction (LV-EF ≤ 35%), an implantable cardioverter defibrillator (ICD) should be implanted, as shown in the WEARIT-II-registry. Our goal was to evaluate the effects of GDMT on LV-recovery and reduction of ICD implantations under protection with a defibrillator vest—depending on the uptitration of GDMT. Methods: 339 consecutive patients between August 2017 and September 2020 with newly diagnosed cardiomyopathy and an EF ≤ 35% were analyzed retrospectively by chart review. All patients were protected by a wearable cardioverter defibrillator (WCD). GDMT as recommended by the ESC started at discharge from hospital. The left ventricular ejection fraction (LV-EF) was determined by transthoracic echocardiography at week 4, 8 and at week 12 (in case of prolonged WCD wear time). Uptitration was performed after 4 and 8 weeks during patient visits. We focused on baseline medication as per GDMT and the dosage increase at week 4, 8 and 12. The aim was the uptitration to the maximum dosage tolerated by the patient. We also compared the LV-EF improvement in the group with and without uptitration of medication dosage. Results: The patient age was, on average, 63.2 years (SD ± 11.9 years). A total of 129 pts (38%) had ICM, 196 (58%) had NICM (incl 66 pts (19%) with DCM and 51 pts (15%) with Myocarditis, 79 pts (24%) with unknown origin) and 14 pts (4%) had other entities (e.g., Tachycardiomyopathy). In total, 21 pts (6%) had an LV-EF of less than 16%, 130 pts (38%) between 16–25% and 183 pts (54%) between 26–35%. GDMT started at discharge from the hospital included treatment with beta blocker for 327 pts (96.5%), ACE-inhibitors/Angiotensin/ARNI for 283 pts (83.5%) and Mineralcorticoid receptor antagonists (MRA) for 334 pts (88.4%). Uptitration was performed in all groups at a rate of 82.3%, 91.1% and 81.0% after 4 weeks and 64.7%, 50.3% and 66.3% after 8 weeks, respectively. After 4 weeks, 25 pts (7.4%) and, after 8 weeks, 171 pts (50.4%) had an EF increase of 5% or more (mean 14.2%). After 4 weeks, 81 patients had an LV-EF more than 35%. A total of 169 pts had a wear time of 12 weeks and an improvement of LVEF of more than 35%. Interestingly, in our study we did not find a significant difference in LV-EF improvement between the group with no uptitration and the group with uptitration. Conclusions: Guideline-directed medical therapy under protection with a WCD from ventricular arrhythmia can reduce the need for implantation of an ICD and can lead to an improvement of ejection fraction. Interestingly, the LV-EF improvement depends on the GDMT at discharge. Current guidelines recommend an initiation of all therapy columns of GDMT (sacubitril/valsartan, ACE-inhibitor/AT1-blocker, mineralcorticoidreceptorblocker, beta blocker) at once and further uptitration to the maximal dosage (ESC Guidelines 2021). A further uptitration of all drugs of GDMT should lead to improvement of LV-EF and consequently to a reduction in ICD implantations.

## 1. Introduction

Guideline Directed Medical Therapy (GDMT) is indicated for patients with heart failure (HF) and reduced left ventricular ejection fraction. GDMT comprises four therapy columns: Renin-angiotensin-aldosterone system inhibitors (RAAS/AT1-receptor blockers (ARB)), Valsartan/Sacubitril (ARNIs), mineralcorticoidreceptorantagonists (MRA) and beta blocker (BB) [1]. Following the recommendations from the last heart failure guideline from 2016, these drugs were applied successively. According to the new ESC guidelines from 2021, these therapy columns in GMDT for patients with heart failure and reduced left ventricular ejection fraction should be initiated as fast as possible and simultaneously, which is contrary to common clinical practice.

Patients with HF with reduced ejection fraction have a higher risk for SCD. Solomon et al., 2005 [2], have shown that this risk in patients after acute myocardial infarction is strongly linked to the reduced EF. The risk for SCD may be only temporary, as the EF can recover through GDMT [3].

The DINAMIT trial [4] showed that the prophylactic implantation of an ICD for patients in the early 6–40 days after myocardial infarction does not lead to a reduction of overall mortality. However, the risk for SCD persists, especially in patients during the early post myocardial period. For improvement of left ventricular pump function, GDMT should be started before an internal defibrillator is considered, which is mentioned in the AHA guidelines for the prevention of SCD. Reduced left ventricular ejection fraction is a predictor for SCD, as shown in the VALIANT trial [5]. During this period, guidelines recommend temporary protection with the WCD [6] while uptitrating GDMT to the target dose recommended by the guidelines.

Optimized medical therapy for improvement of left ventricular systolic function is needed in order to prevent remodeling processes of the myocardium. This might result in an LV-EF improvement. Guidelines recommend medical treatment for 3 months before making the decision for an implantable cardioverter defibrillator (ESC guidelines 2016).

In the ESC guidelines for heart failure therapy from 2021, it is emphasized that optimized medical therapy should be started as fast as possible when heart failure with reduced ejection fraction is diagnosed. Furthermore, all therapy columns of GDMT should be applied at once.

The wearable cardioverter defibrillator (WCD) according to ESC guidelines from 2016 provides a temporary protection against ventricular arrhythmias. It also has several diagnostic features that enable the treating health care professional to continuously monitor several vital parameters such as heart rate (HR), activity and body position [7].The purpose of our analysis was to clarify if patients with initiated GDMT at discharge and further uptitration under protection with a WCD have: 1. reduced probability for the need of an implantable cardioverter defibrillator (ICD) and 2. higher LVEF-recovery rates—compared to patients without uptitration of GDMT.

## 2. Methods

In a single center clinical, a retrospective study in the Kerckhoff Heart center Bad Nauheim 339 consecutive patients between August 2017 and September 2020 with newly diagnosed cardiomyopathy and an EF ≤ 35% were protected against SCD with a wearable cardioverter defibrillator (WCD). Further differentiation of the underlying form of cardiomyopathy was conducted before by cardiac MRI or coronary angiography. The study was performed according to good clinical practice guidelines.

The WCD will deliver a defibrillation as per the programmed settings [8].

For all patients, GDMT started at baseline at discharge from hospital. 

GDMT consists of the following substances:ACE/AT1-inhibitors or ARNI;Beta blockers;Aldosteronantagonists.

After initiation of GDMT, uptitration is performed in order to aim for the target dose (maximum dosage of the prescribed drug defined by AHA guidelines [9], Table 1).

All patients were protected with the WCD. The WCD monitors the patient’s heart rate by four dry electrodes that are positioned on the patient’s chest. In case of ventricular arrhythmias, the device issues an acoustic alarm to notify the patient of an imminent defibrillation. If the patient is still conscious, the sequence can be manually stopped by pressing two buttons on the device. If the patient has lost consciousness due to the arrhythmia, the device delivers a shock therapy.

As part of standard clinical care, re-evaluation of the left ventricular ejection fraction (LV-EF) was carried out by transthoracic echocardiography in our hospital at week 4, 8 and 12. Uptitration of GDMT was performed under hospital surveillance after 4 and 8 weeks. In our study, LV-EF improvement is considered after an increase of more than 5% compared to the initial value of LV-EF. If the LV-EF was improved to a value of more than 35% after 4 or 8 weeks, the WCD was no longer indicated and the wear time was stopped. If there was no improvement after 8 weeks under GDMT, the WCD wear time was prolonged until planned ICD implantation (between 3 and 4 weeks after 8 weeks follow-up). So, this means an ICD was implanted (in case of an LV-EF under 35% at time of implantation) after 12 weeks, which corresponds to the guidelines. If there was an improvement of LV-EF after 8 weeks, but the value is less than 35%, the wearing time of the WCD was extended for another 4 weeks. Afterwards, the LV-EF was re-evaluated with transthoracic echocardiography. If, then, the LV-EF was still under 35%, the patient was supplied with a defibrillator.

Uptitration is associated with an increase in dosage of each drug included in GDMT. Uptitration was only performed if the patient tolerates the medication and the blood pressure was stable. That means the patients did not develop any symptoms of hypotension. Regarding ACE-inhibitors or AT-receptor blockers, we always tried to upgrade to sacubitril/valsartan after 4 weeks if there were no contraindications. We analyzed baseline medication as per whether GDMT was started or not. Afterwards, we looked to see if GDMT was uptitrated or not. Another important part of the analysis was how far uptitration of GDMT was possible to the target dose after 4 and 8 weeks.

Uptitration was described by the percentage of patients receiving any increase in medication dosage and the percentage of maximal medication dosage tolerated by patients compared to the target dose. It was also an important issue to know if uptitration of GDMT was possible after 4 and 8 weeks or if there were differences noticeable between these different timepoints.

We focused on descriptive statistics by calculating the mean values of LV-EF improvement, the distribution of cardiomyopathy forms, the adherence of patients GDMT, the percentage of any medication dosage increase (uptitration) performed and the percentage of ICD implantations. Adherence to GDMT means, in that case, whether patients took the prescribed medication in the defined dosage. If malignant arrhythmia was detected by the WCD, a shock therapy was applied. A comparison group where patients did not wear a WCD was not included because our focus was on the effect of GDMT and its uptitration in patients with heart failure with reduced ejection fraction.

## 3. Results

The patient characterics are summarized in Table 2. The average patient age was 63.2 years (SD ± 11.9 years), and 19% of the patients were female. The patients were categorized regarding the underlying form of cardiomyopathy (ischemic, nonischemic and others such as tachycardiomyopathy). A total of 129 pts (38%) had ischemic cardiomyopathy (ICM), 196 (58%) had nonischemic cardiomyopathy (NICM) (incl 66 pts (19%) with DCM, 51 pts (15%) with Myocarditis and 79 pts (24%) with unknown origin) and 14 pts (4%) had other entities (e.g., Tachycardiomyopathy). In total, 21 pts (6%) had an LV-EF of less than 16%, 130 pts (38%) between 16–25% and 183 pts (54%) between 26–35%. 

In five patients, the LV-EF was unknown, because these patients were supplied with a WCD after device explanation or for secondary prevention after ventricular tachycardia.

The average wear time of the WCD was 56 d (±36.8). The compliance during each day worn was 23.44 h (±4.1).

Baseline medication started in hospital included treatment with Beta blocker (BB) for 327 pts (96.5%), ACE-inhibitors/Angiotensin-1-inhbitors/ARNI for 283 pts (83.5%) and Mineralcorticoid receptor antagonists (MRA) for 334 pts (88.4%) (Figure 1). Uptitration has occurred in all medication groups in 82.3% (BB), 91.1% (ACE-inhibitor/Angiotensin-1-inhibitors/ARNI) and 81% (MRA) after 4 weeks and 64.7% (BB), 50.3% (ACE-inhibitors/Angiotensin-1-inhibitors/ARNI) and 66.3% (MRA) after 8 weeks of all patients, respectively (Figure 2).

The percentage of uptitration was, after 8 weeks, lower than after 4 weeks. After 4 weeks, 25 pts (7.4%) and, after 8 weeks, 171 pts (50.4%) had an EF increase of 5% or more (mean 14.2%). The median EF increase was 13 % (Figure 3). NICM patients had a slightly higher increase in average EF than patients with ICM (14.6% and 13.3%, respectively), and 109 pts (32.1%) received an implantable Cardioverter Defibrillator (ICD) because of persistently reduced systolic LV-EF < 35%, whereas 230 pts (67.8%) did not receive a device (Figure 4).

An improvement of LV function of more than 35% can be detected in 81 (24%) patients after 4 weeks and 65 pts after 8 weeks. A total of 169 patients had an LV function above 35% after 12 weeks (Figure 5). In total, 109 patients had to be supplied with an ICD.

The rate of ICD recipients was slightly higher in the group that did not receive uptitration. This trend was not statistically significant (65.1% vs. 67.4%).

## 4. Discussion

With the WCD, patients with increased risk for SCD can be protected transiently (e.g., for medical heart failure therapy) from ventricular arrhythmias. Implementation and uptitration of GDMT reduces the need for ICD implantation due to improvement of LV-EF. As shown in the results, there were not any differences noticed between different forms of cardiomyopathy (NICM, ICM or other).

Patients benefit from a high percentage of GDMT at discharge. This is recommended in the new ESC guidelines from 2021, where all drugs should be started at once. In the past, GDMT was applied sequentially. The rate of ACE-inhibitor/ARNI is lower compared to beta blocker and MRA. A reason for that fact could be the lack of hemodyamic tolerance at the beginning especially for Entresto.

Due to an already high percentage of GDMT at discharge, it was more difficult to achieve an uptitration after 8 weeks compared to 4 weeks. This led to a decrease in the percentage of uptitration of medication.

Our study has shown that implementation of GDMT and the early uptitration under clinical settings can improve LV-EF and consequently reduce the number of ICD implantations. The clinical appointments after 4 and 8 weeks helped in general to adjust the GDMT and titrate it to the optimal dosage tolerable for the patients. This kind of heart failure therapy with close follow-ups after discharge is addressed, for example, by Roth et al. [10]. Their study has demonstrated that the performance and right timing of postdischarge follow-ups can lead to a reduction of hospitalizations. Heart failure patients after discharge are exposed to the risk of cardiac decompensation leading to hospitalization. For these patients, there is no uptitration of GDMT performed. This issue is addressed, for example, by Greene et al. [11], where patients with low rates of GDMT who also lack of uptitration have significantly reduced 1-year survival compared to patients with adequate GDMT. 

The ESC guidelines from 2021 give a Class IC recommendation based on evidence-medicine for GDMT after discharge and optimizing it at follow-up visits [12].

Unfortunately, uptitration of all medications to the target dosage was not possible in all patients, which can be seen in Figure 2. A reason could be that a high rate of GDMT was already initiated at discharge and a high rate of uptitration was already performed after 4 weeks. For this reason, further uptitration after 8 weeks could be only achieved in a lower number of patients, because a higher dose was not tolerated hemodynamically by all patients. Implementation of GDMT at time of discharge and the further uptitration under clinical surveillance is an important therapeutical approach in LV-EF improvement, which is an important recommendation in the ESC guidelines in medical heart failure therapy from 2021.

The CHAMP-HF registry underlines the positive association of initiation and uptitration of GDMT and reduction of heart failure events [11]. 

In our study, we were not able to find any significant difference in LV-EF improvement between the group in which GDMT was uptitrated at week 4 and 8 and the group with no further GDMT uptitration. We only found a slight trend noticeable after uptitration, which may depend on the high number of GDMT at the timepoint of discharge from hospital. After 2 months, the number of primary prophylactic ICD indications is reduced to a third. This amount of ICD implantations is less compared to the PROLONG study by Duncker et al. [13]. In the PROLONG study, 62% of patients with cardiomyopathy had an improvement of LVEF under GDMT and did not receive an implantable cardioverter defibrillator. In our patient cohort, 67.8% patients did not receive an implantable cardioverter defibrillator.

The WCD is a useful tool to protect patients from malignant ventricular arrhythmia. Fortunately, we had in our cohort no ventricular tachycardia terminated by shocks in the Vest trial [14]. In total, 1.4% of the patients received an appropriate shock therapy from the defibrillatory vest. We assume that implementation and uptitration had an important impact for avoiding shock therapy in our cohort due to LVEF improvement. Especially, the use of Entresto, as shown in different trials, leads to the reduction of ventricular arrhythmias [15].

Our results give a possible hint that the LV-EF improvement could depend on a high rate of GMDT at discharge and further uptitration after discharge. Gracia et al. [16] also showed in their study that the implementation of GDMT at discharge improves the outcome of patients with heart failure. Delaying GDMT leads to worsening of HF progression.

## 5. Conclusions

Implementation and uptitration of GDMT with close ambulatory follow-ups (4 and 8 weeks) for patients with heart failure with reduced ejection fraction (EF < 35%) under protection with a WCD from ventricular arrhythmia can reduce the need for the implantation of an ICD. An improvement of LV function to more than 35% can be detected in nearly half of the patients.

The surveillance of these patients after discharge and adjustment of GDMT over 2 months allowed us the reduction of ICD implantations due to improvement of left ventricular function to a significant low rate. This effect could be also observed in other clinical trials (Dunker et al. and Gracia et al.). However, compared to them we could show that a high percentage of optimized medical therapy at discharge and its uptitration in the hospital has a positive effect regarding the reduction of ICD implantations after the waiting period.

As emphasized in the current ESC guidelines from 2021, it is important to establish all pillars of GDMT at once and not sequentially.

In our study, there was no significant difference if uptitration of medication was performed or not, but there was a slight trend noticeable in favor of patients experiencing uptitration of GDMT.

All in all, WCD and, more importantly, initiation and uptitration of GDMT, can prevent patients with heart failure from ICD implantations and lead to improvement of LV-EF.

### Study Limitations

This analysis was retrospective. It is not clear if dosage change was only conducted in the hospital or by a general practitioner between the hospital visits after 4 and 8 weeks. Moreover, it is not possible to describe the patient’s compliance with optimized medical therapy. We could not find all reasons why some patients had no uptitration of GDMT. Medication compatibility can differ among patients from the study.

## Figures and Tables

**Figure 1 jcm-11-06122-f001:**
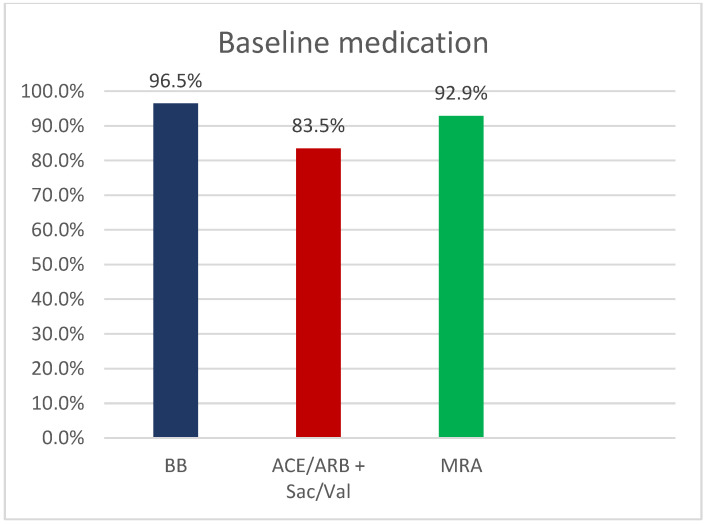
Baseline medication at discharge. BB = Betablocker, ACE = ACE-Inhibitor, ARB = Angiotensin-receptor blocker, MRA = Mineralcorticoidreceptorantagonists, Sac/Val = Sacubitril/Valsartan.

**Figure 2 jcm-11-06122-f002:**
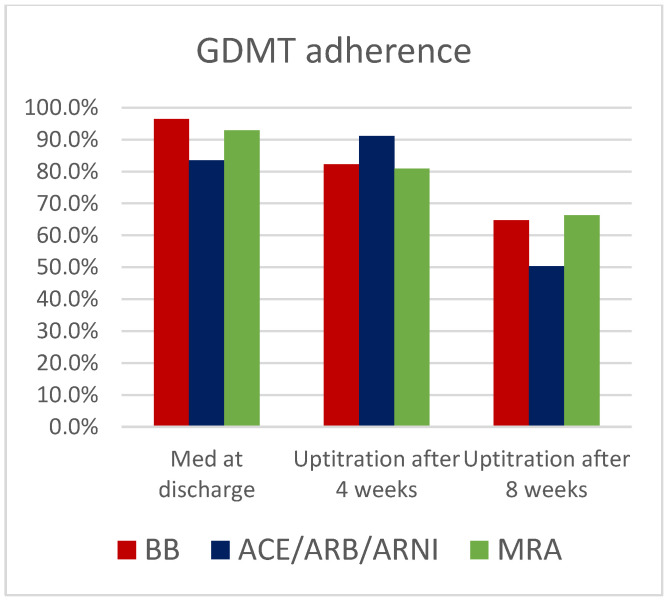
Rate of GDMT adherence and uptitration at discharge, after 4 and 8 weeks.

**Figure 3 jcm-11-06122-f003:**
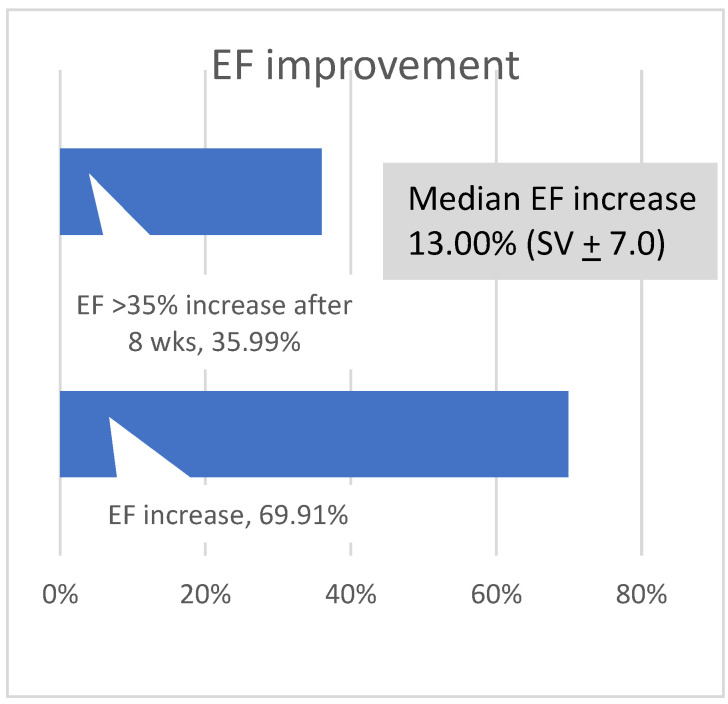
LV-EF improvement after 8 weeks.

**Figure 4 jcm-11-06122-f004:**
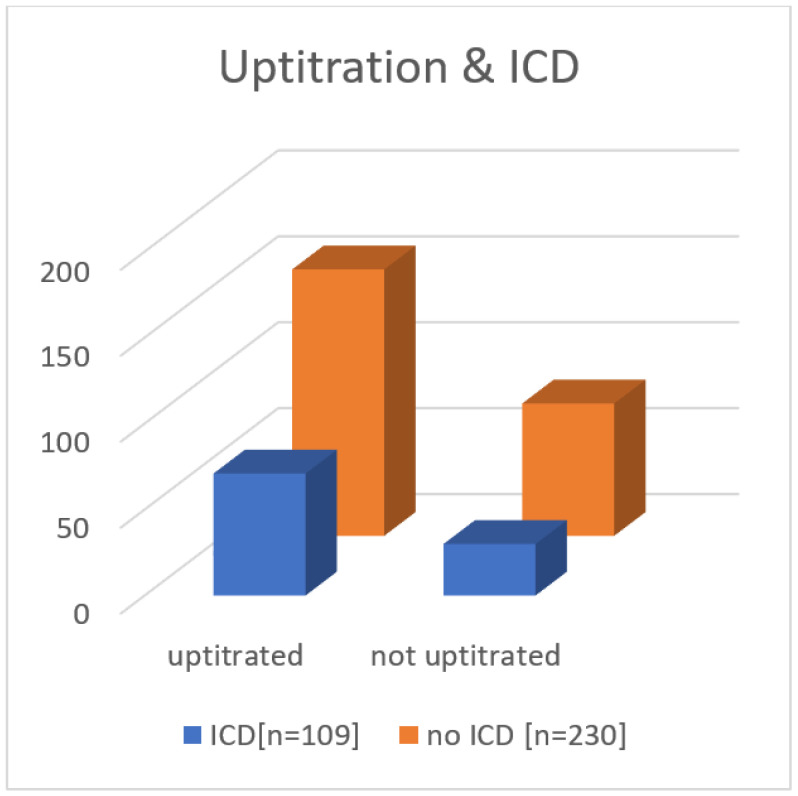
Number of ICD implantations after uptitration and without uptitration (uptitration = any increase in medication dosage). ICD = Implantable cardioverter defibrilator.

**Figure 5 jcm-11-06122-f005:**
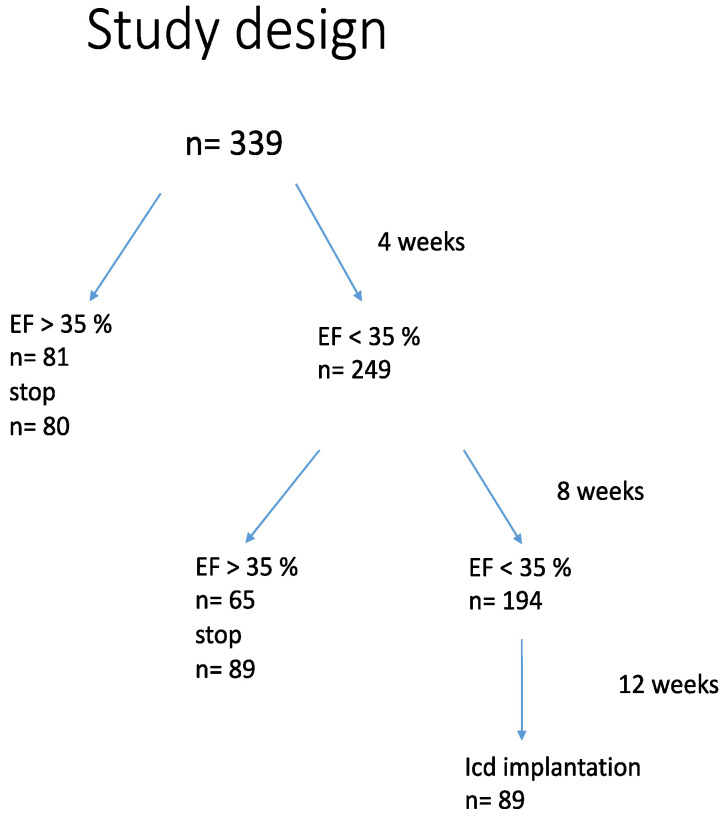
Study design of WCD wear time and time point for ICD implantation.

**Table 1 jcm-11-06122-t001:** Drugs of GDMT and their target dose defined by AHA guidelines in 2017 (Yancy et al.).

Drug	Initial Daily Dose (mg)	Target Dose (mg)
ACE inhibitors		
Captopril	6.25 mg	50
Enalapril	2.5	20
Fosinopril	5	40
Lisinopril	2.5	20
Perindopril	2	16
Quinapril	5	20
Ramipril	1.25	10
Trandolapril	1	4
ARBs		
Candesartan	4	32
Losartan	25	150
Valsartan	20	160
ARNI		
Sacubitril/valsartan	24/26	97/103
Aldosterone antagonists		
Spironolactone	12.5	25
Eplerenone	25	50
	Beta blockers	
Bisoprolol	1.25	10
Metoprolol	25	200
Nebivolol	1.25	10
Carvedilol	12.5	50

**Table 2 jcm-11-06122-t002:** Patient characteristics.

Parameter	Category	Value
**Gender**	male	*n* = 275 (81%)
	female	*n* = 64 (19%)
**Form of Cardiomyopathy**	NICM	*n* = 196 (58%)
	ICM	*n* = 129 (38%)
	other indications (e.g., tachycardiomyopathy)	*n* = 14 (4%)
Age	average	63.2 years (SD ± 11.9 years)
**LV-EF distribution** (at discharge)	<16%	*n* = 21 (6%)
	16–25%	*n* = 130 (38%)
	26–35%	*n* = 183 (54%)

## Data Availability

Data used in this study can be made available upon reasonable request.

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
