# Peer review of "Guideline Directed Medical Therapy at Discharge and Further Uptitration Leading to Reduction in Indication for Prophylactic ICD Implantation during Protected Waiting Period"

_jcm, 2022, doi:10.3390/jcm11206122_

Round 1

Reviewer 1 Report

I completely agree with your opinion that GDMT improved LV-EF and avoidance of prophylactic ICD implantaion. Howevere this is widely known, it seems to be difficult to find new clinical implication in this article. If you could show us that your recommendation of WCD and ICD implantation in HFrEF patients according to this data, your article would have more meanings.

And, I have some questions about this article

1. Is it appropriate to decide to perform ICD implantation at 8weeks in Non-EF improvement group? To say it is ture, you should show that there was no imrpvement in LV-EF after ICD implantation in this group.

2.I'd like to know whether ICD implanted patient have appropriate ICD shocks or not.

3.Do non-ICD implanted patient have any arrithmic event ?

4.I can't understand the meaning of Graph 3. It means that medication dosage decrease at 8 weeks than 4 weeks ? 

In addition, I found mistakes as follows, please revise.

P3 If the patient fas lost consciousness due to the arrhythmia, the device. →Sentence is cut off in the middle.

P8 Thi amount of icd implantations is less compared to the PROLONG study by[15].→Sentence is cut off in the middle. And you should write ICD.

Author Response

Question 1:

Total follow up time was 3 month with clinical follow ups after 4, 8 and 12 weeks. In patients without any improvement after 2 month we decided to implant an icd after 2 month. Between the 8 weeks follow up and the icd-implantation we had another waiting period of 4 weeks in which we did not found further LV-EF improvement in this group. The icd implantation in this group was done on average 12 weeks after starting optimized GDMT. We rewrote this in the methods section.

Question 2:

Our primary foucs was the effect of GDMT in EF improvement under WCD. The effect on patients after ICD implanattion is not part of our analysis, so we did not analyse the rate of icd therapies after icd implantation.

Question 3:

In our analysed group there were no arrhythmic events that led to shock therapy via life vest during wearing period.

Question 4:

Graph 3 shows the adherence at discharge and the rate of uptitration – not the dosage of each medication ist shown at week 4 and 8.The rate of uptitration (increase of medication dosage) decreases after 4 and 8 weeks, because at the beginning patients had already a high rate of GDMT (more than 80 %) – following the gap of reaching the end dosis was low

Question 5:

The two sentences in page 3 and 8  were corrected.

Reviewer 2 Report

Congratulations to the authors for the very interesting idea of the manuscript. Page 3 of 9 “If the patient has lost consciousness due to the arrhythmia, the device.” The phrase Stops without any reason. The phrase: “Heart failure patients after discharge is exposed to the risk to get events of cardiac decompensation leading to hospitalization and that uptitration of GDMT is not performed under ambulatory conditions “ is not of immediate understanting.

Author Response

The mistake was corrected in the revised version.

Reviewer 3 Report

The topic is interesting. However it is not clear the aim of the study, sine international guidelines recommend LVEF re-evaluation after 3 months of GDMT, hilighting the very low arrhythmic risk in this period.

The abstract should be reduced and without refs.

Intro: The 4 columns of HFrEF therapy include also SGLT2i!!! Non data about SGLT2i in the results.

Table 1: carvedilol and nebivolol are lacking.

It is uncorrect to include pts implanted for secondary prevention!

Author Response

Question 1:

Total follow up time was 3 month with clinical follow ups after 4, 8 and 12 weeks. In patients without any improvement after 2 month we decided to implant an icd after 2 month. Between the 8 weeks follow up and the icd-implantation we had another waiting period of 4 weeks in which we did not found further LV-EF improvement in this group. The icd implantation in this group was done on average 12 weeks after starting optimized GDMT. We rewrote this in the methods section.

Question 2:

The abstract was reduced and the refs removed.

Question 3:

Our data collection was done before including SGLTIIi into the HF guidelines (2017-09/2020)

Question 4:

Nebivolol and carvedilol  were now added in the table

Question 5:

Thats correct. Among 5 patients the LV-EF was unknown, because these patients were supplied with a WCD after device explantation or for secondary prevention after ventricular tachycardia. These patients were excluded from the study.

Round 2

Reviewer 1 Report

Thank you for answering my questions.

I've understood the point that you'd like to emphasis. As you mentioned in this article, I recomfirmed that GDMT is important to improve LV function and avoid unnessesary ICD implantation. 

I'm interested in the long term prognosis of the patients who were included in this study. I look forword to seeing the data in your another article. 

Additional comments,

I think the two following sentences are imcomplete. Please revise to complete sentences.

P3 : If the patient has lost consciousness due to the arrhythmia, the device.

P8 : This amount of ICD implantations is less compared to the PROLONG study by

Author Response

Thank you very much for the review. 

The sentences are now corrected in pages 3 and 8.

Reviewer 3 Report

Nothing to add

Author Response

Thank you very much for the review.